# The Intrinsic GDP/GTP Exchange Activities of Cdc42 and Rac1 Are Critical Determinants for Their Specific Effects on Mobilization of the Actin Filament System

**DOI:** 10.3390/cells8070759

**Published:** 2019-07-21

**Authors:** Pontus Aspenström

**Affiliations:** Rudbeck Laboratory, Department of Immunology, Genetics and Pathology (IGP), Uppsala University, SE 751 85 Uppsala, Sweden; pontus.aspenstrom@igp.uu.se; Tel.: +46-18-4710000

**Keywords:** fast-cycling, Rho GTPases, Cdc42, Rac, filopodia, actin

## Abstract

The Rho GTPases comprise a subfamily of the Ras superfamily of small GTPases. Their importance in regulation of cell morphology and cell migration is well characterized. According to the prevailing paradigm, Cdc42 regulates the formation of filopodia, Rac1 regulates the formation of lamellipodia, and RhoA triggers the assembly of focal adhesions. However, this scheme is clearly an oversimplification, as the Rho subfamily encompasses 20 members with diverse effects on a number of vital cellular processes, including cytoskeletal dynamics and cell proliferation, migration, and invasion. This article highlights the importance of the catalytic activities of the classical Rho GTPases Cdc42 and Rac1, in terms of their specific effects on the dynamic reorganization of the actin filament system. GTPase-deficient mutants of Cdc42 and Rac1 trigger the formation of broad lamellipodia and stress fibers, and fast-cycling mutations trigger filopodia formation and stress fiber dissolution. The filopodia response requires the involvement of the formin family of actin nucleation promotors. In contrast, the formation of broad lamellipodia induced by GTPase-deficient Cdc42 and Rac1 is mediated through Arp2/3-dependent actin nucleation.

## 1. Introduction

The function of the Rho GTPases was initially unraveled about 25 years ago in a series of seminal papers from the laboratory of Alan Hall [1,2,3]. The concept was established that each of the three GTPases Cdc42, Rac1, and RhoA have specific effects on the actin filament system, and, thereby, on the morphology and the migratory properties of cells. According to the resulting paradigm, Cdc42 regulates the formation of filopodia and Rac1, the formation of lamellipodia, and RhoA is involved in the assembly of stress fibers. The most simplistic consequence of this concept is that cell migration requires the concerted activity of only these three members of the Rho GTPases. However, we now know that mammalian genomes express 20 different members of the Rho GTPases that have more diverse effects on the organization of the actin filament system than originally believed [4].

The Rho GTPases represent a subfamily of the large Ras superfamily of small GTPases [5]. In many ways, they behave and are regulated in similar ways to the Ras GTPases, because they cycle between their inactive GDP-bound and active GTP-bound conformations [6]. This cycling is regulated by proteins that can trigger the replacement of GDP for GTP (the guanine nucleotide exchange factors, GEFs) and proteins that can catalyze the GTP hydrolysis (the GTPase activating proteins, GAPs) [7,8]. In addition, a third group of proteins is involved in the regulation of the activity of Rho GTPases, the guanine nucleotide dissociation inhibitors (RhoGDIs) [9]. The RhoGDIs sequester Rho GTPases in an inactive complex, and upon cell stimulation, the complex dissociates to release the Rho GTPases, which can then be activated by the GEFs.

Importantly, the Rho GTPases can be further subdivided into the classical and atypical Rho GTPases, whereby this mode of regulation by RhoGDIs, GEFs, and GAPs is only applicable to the classical Rho GTPases. There are now 10 members of the classical Rho GTPases: Rac1, Rac2, Rac3, RhoG, Cdc42, TCL, TC10, RhoA, RhoB, and RhoC. The atypical Rho GTPases, in turn, fall into two categories: those that are GTPase-deficient, and those that are fast-cycling Rho GTPases. The first of these categories includes Rnd1, Rnd2, Rnd3/RhoE, RhoH, RhoBTB1, and RhoBTB2 [10,11]. These proteins lack, or have very low, intrinsic GTP hydrolysis activity, so in essence, they are constitutively GTP bound. The fast-cycling Rho GTPases include RhoD, RhoF, Wrch-1 (RhoU), and Chp (RhoV), and these have greatly elevated intrinsic GDP/GTP exchange activity [11].

Inside cells, the concentration of GTP is 10-fold higher than that of GDP, and, therefore, in 9 out of 10 cases, the fast-cycling Rho GTPases will have GTP in their active site. As a result, they will remain predominantly in their active, GTP-bound, conformation [12]. Thus, the Rho GTPases are not a homogenous group of proteins, and along with Cdc42, Rac1, and RhoA, there are 17 other members of the Rho family. These other Rho GTPases are also generally expressed ubiquitously, with the exception of RhoH and Rac2, which appear to be mainly expressed in hematopoietic cells [13].

Another factor that must be taken into account in the regulation of the signaling potency of the Rho GTPases is the degree of GDP/GTP exchange activity of these Rho GTPases, i.e., how fast they can cycle between their GDP-bound and GTP-bound conformations. A large number of studies on Cdc42 and Rac1 have used the G12V and Q61L mutants of these GTPases. These mutations are known to block the hydrolysis activities of the small GTPases [14]. In this regard, another set of mutations appear to be more appropriate for consideration as constitutively active mutants: the fast-cycling mutants of Cdc42 and Rac1, or more specifically, the Cdc42/F28L and Rac1/F28L mutants [15,16]. These mutant Rho GTPases were designed based on earlier studies on Ha-Ras, where it was shown that Ha-Ras/F28L has a lower affinity for the guanine base in the GTP binding pocket in Ha-Ras, which results in elevated GDP/GTP exchange activity, i.e., a fast-cycling phenotype [17]. Another example is the splice variant of Rac1, known as Rac1B, which was originally identified in colorectal and breast cancer [18,19]. Rac1B is a result of an alternative splicing event that provides 19 extra amino-acids following amino-acid 75, immediately after the Switch-II motif (18,19). Importantly, the Rac1B variant turned out to have much higher intrinsic GDP/GTP exchange activity and might, therefore, qualify as a fast-cycling mutant protein [20,21,22].

In the present study, the cellular effects elicited by GTPase-deficient and fast-cycling mutants of Cdc42 and Rac1 are compared. The GTPase-deficient mutants of Cdc42 and Rac1 are shown to predominantly trigger the formation of broad lamellipodia and the assembly of stress fibers. In contrast, the fast-cycling mutants of these two Rho GTPases trigger the formation of filopodia, accompanied by the dissolution of stress fibers.

## 2. Experimental

### 2.1. Mutant Small GTPases

The Cdc42 and Rac1 mutations analyzed in this study are presented in Table 1. Myc epitope-tagged wild-type Cdc42 (Cdc42/wt) and Cdc42/Q61L, Cdc42/G12V, Cdc42/T17N, Rac1/wt, Rac1/Q61L, and Rac1/T17N were described previously [4]. Cdc42/Q61LF37A and Cdc42/Q61LY40C were generous gifts from Alan Hall (University College London, UK). Rac1/P29S was constructed by GeneScript Biotech, Leiden, The Netherlands. All of the other mutants of Cdc42 and Rac1 were generated by the QuickChange protocol (Stratagene) according to the procedures provided by the manufacturer. See Table 1A for an overview of the biological function of the mutated amino acid residues.

### 2.2. Antibodies and Reagents

Rabbit polyclonal anti-Myc antibodies and Tetramethyl rhodamine isothiocyanate (TRITC)-conjugated phalloidin were from Sigma. Alexa Fluor 488-conjugated anti-rabbit antibodies were from Molecular Probes. The inhibitors used in the study are described in Table 1B, and were from Sigma.

### 2.3. Cell Cultivation and Transfection

BJ/hTERTSV40T human foreskin fibroblasts were cultured in Dulbecco’s modified Eagles medium (Gibco) supplemented with 10% fetal bovine serum (FBS, Sigma-Aldrich, St Louis, MO, USA) and 1% penicillin/streptomycin (from a stock solution of 5000 U/mL penicillin and 5000 μg/mL streptomycin from Gibco, ThermoFisher, Waltham, MA, USA). Porcine aortic endothelial cells stably transfected with the human PDGF β-receptor, known as PAE/PDGFRβ cells (clone β1:3), were cultured in Ham’s F12 medium (Sigma), supplemented with 10% FBS and 1% penicillin/streptomycin. All of these cells were cultured at 37 °C in an atmosphere of 5% CO_2_.

The BJ/hTERTSV40T cells were transfected using JetPEI (Polyplus Transfection, Illkirch, France), and the PAE/PDGFRβ cells were transfected using Lipofectamine (Invitrogen Life Technologies, Carlsbad, CA, USA), using the protocols provided by the manufacturers.

For immunocytochemistry, the coverslips were fixed in 3% paraformaldehyde in phosphate buffered saline (PBS) for 20 min at 37 °C, and then washed with PBS. The cells were then permeabilized in 0.2% Triton X-100 in PBS for 5 min, washed again in PBS, and then incubated in 5% FBS in PBS for 30 min at room temperature. The primary and secondary antibodies were diluted in PBS containing 5% FBS. The cells were incubated with the primary antibodies, followed by the secondary antibodies, for intervals of 1 h, with a washing step in-between. The coverslips were mounted on object slides using Fluoromount-G (Southern Biotechnology Associates, Birmingham, AL, USA). The cells were photographed using a digital camera (Zeiss AxioCAM MRm, Oberkochen, Germany) connected to a microscope (Zeiss AxioVert 40 CFL, Oberkochen, Germany), with the images processed using the Zeiss AxioVision software (Zeiss, Oberkochen, Germany).

For the analysis of cells morphologies, transfected cells were analyzed by microscopy and scored for the formation of filopodia, broad lamellipodia, broad stress fibers, and stress fiber dissolution, according to the criteria demonstrated in Appendix A. In order to validate the method, a set of blinded slides were quantified, and comparable results were obtained (data not shown). It can sometimes be difficult to distinguish filopodia from retraction fibers. Live cell imaging is often used to identify actively growing protrusions, but this method is not suitable for large scale analysis of several hundred transfected cells for each condition. Retraction fibers are often seen during cell rounding up and retraction fibers often leave traces of material behind. Cells expressing these phenotypes were not included in the analysis. Importantly, cells with equal apparent expression of the various Cdc42 and Rac1 mutants were selected for the analysis (see examples in Appendix A). Cells with too high or too low expression were excluded from the analysis. In addition, only cells with intact nuclei were subjected to quantifications (Appendix A). Unpaired two-way Student’s *t*-tests with unequal variance were performed to calculate statistical significance for the data in Figure 3D–F. One-way ANOVA with Tukey’s post hoc analyses were made to calculate statistical significance for the data in Figure 1B,C, Figure 2B,C, Figure 3B,C and Figure 4B,C using the GraphPad Prism software. The complete result is presented in Appendix A.

For the analysis of cell shape shown in Figure 3D–F, 20 images of transfected cells and mock-transfected cells (treated with JetPEI without DNA) per condition were analyzed for circularity, cell perimeter, and cell area using ImageJ.

## 3. Results

### 3.1. An Intact GDP/GTP Exchange Activity is the Basis for Cdc42-Induced Filopodia Formation

We have previously shown that the Cdc42/Q61L so-called constitutively active mutant of Cdc42 induces the formation of lamellipodia and thick stress fibers in PAE/PDFGRβ cells [4]. This result is in apparent contradiction to the current paradigm, which states that Cdc42 is specifically involved in the formation of filopodia [23]. The common explanation for this Cdc42-induced lamellipodia formation is that Cdc42 activates Rac1. This concept is based on the observation of Nobes et al. (1995) that constitutively active Cdc42/G12V needed to be co-injected with a dominant-negative Rac1 mutant to promote formation of filopodia in Swiss 3T3 fibroblasts [3,24]. Another explanation that does not necessarily exclude the possibility of an involvement of Rac1 relates to the intrinsic enzymatic properties of the Cdc42 mutants used. The commonly used constitutively active Cdc42 mutants, Cdc42/G12V and Cdc42/Q61L, are GTPase-deficient, which means that they are locked in the GTP-bound conformation [14]. Another set of Cdc42 mutants, as represented by Cdc42/F28L, have been shown to have higher intrinsic GDP/GTP exchange activities [15,16]. To compare the effects on actin dynamics elicited by these two categories of Cdc42 mutants, BJ/hTERTSV40T fibroblasts were transiently transfected with plasmids encoding Cdc42/wt, Cdc42/Q61L, Cdc42/F28L, and the dominant-negative Cdc42/T17N mutant. In agreement with previous observations, Cdc42/Q61L induced the formation of broad lamellipodia and the assembly of broad stress fibers in 55.6 ± 11.8% and 90.1 ± 1.0% of the cells, respectively (Figure 1A–C) [4]. The lamellipodia are much broader in these Cdc42/Q61L-expressing cells than the normal lamellipodia seen in mock-transfected fibroblasts, and the stress fibers also appear broader and more spread out compared to the mock-transfected fibroblasts (Figure 1A, see Appendix A for description of the criteria for these quantifications). Only 18.9 ± 5.2% of the Cdc42/Q61L-expressing cells had filopodia. In contrast, the Cdc42 variants that can still cycle between their GDP-bound and GTP-bound conformations, i.e., Cdc42wt and Cdc42/F28L, induced the formation of filopodia in 78.4 ± 8.9% and 61.9 ± 3.1% of the transfected cells, respectively (Figure 1A–C, for the calculated values of statistical significances, see Appendix A). Moreover, expression of Cdc42/wt and Cdc42/F28L resulted in robust dissolution of stress fibers in 84.0 ± 1.8% and 54.0 ± 12.1% of the transfected cells, respectively. Similar responses were triggered by the different Cdc42 variants when expressed in porcine aortic endothelial (PAE/PDGFRβ) cells (Appendix A). Two additional mutations were analyzed here: Cdc42/G12V and Cdc42/D118N. Cdc42/G12V is a classical GTPase-deficient ‘constitutively active’ mutant, and it induced formation of broad lamellipodia in 38.1 ± 16.2% of the cells, filopodia in 35.3 ± 5.9% of the cells, and broad stress fibers in 62.1 ± 10.5% of the cells, i.e., the balance is shifted more towards filopodia formation compared to Cdc42/Q61L (Figure 2A–C, Appendix A). Cdc42/D118N has been described as exchanging GDP for GTP more rapidly than wild-type Cdc42, but significantly more slowly than the Cdc42/F28L mutant [25]. Therefore, it was anticipated that Cdc42/D118N would give rise to cellular responses similar to Cdc42/F28L. In should be noted that these measurements were performed in vitro with the recombinant proteins, and it is likely that the kinetic properties of Cdc42/D118N are different in vivo. It was found that BJ/hTERT SV40T cells expressing Cdc42/D118N resembled the Cdc42/Q61L-expressing cells (19.3 ± 5.5% with filopodia, 54.7 ± 6.7% with lamellipodia, 73.0 ± 7.5% with broad stress fibers, where 24.3 ± 5.0% had lost the stress fibers, Figure 2A–C).

### 3.2. The Role of the Effector Loop in Cdc42-Induced Filopodia Formation

Previous studies have shown that mutations in codon 37 in the effector loop of Rac1 abolish the lamellipodia response, whereas mutations in codon 40 do not inhibit lamellipodia formation [26]. The equivalent mutants of Cdc42 were shown to bind to the same spectrum of effectors, although both of these Cdc42 effector-loop mutants triggered filopodia formation. To study the involvement of the effector loop in Cdc42-dependent cytoskeletal reorganization, three effector loop mutations in a Q61L background were transiently transfected in the BJ/hTERTSV40T fibroblasts: Cdc42/Q61LY40C, Cdc42/Q61LF37A, and Cdc42/Q61LT35A. In line with previous results, Cdc42/Q61LY40C retained the Cdc42/Q61L-induced formation of broad lamellipodia (in 55.5 ± 11.8% of the cells) and broad stress fibers (in 70.7 ± 2.3% of the cells) (Figure 2A). Expression of Cdc42/Q61LF37A induced the formation of broad lamellipodia in significantly fewer cells (13.9 ± 2.6%), and instead, filopodia formation was triggered in 39.3 ± 3.3% of the cells. However, in contrast to Cdc42/F28L, expression of the Q61LF37A mutant did not result in stress fiber dissolution (Figure 2A–C, Appendix A). Interestingly, mutation of the threonine in position 35 to an alanine in the Cdc42/Q61 background suppressed the cellular effects seen with Cdc42/Q61L. Instead, expression of Cdc42/Q61LT35A triggered the formation of filopodia in 62.4 ± 5.8% of the cells and resulted in stress fiber dissolution in 42.3 ± 2.8% of the cells (Figure 2A–C). The so-called insert domain that encompasses amino acids 120 to 134 of Rac1 and Cdc42 is present in all members of the Rho family of small GTPases and has been shown to provide an additional effector-binding interface [13,27]. The insert domain, as amino acids 120–134, was removed in Cdc42/Q61L to create a Cdc42/Q61LΔins mutant. Expression of this deletion mutant in the BJ/hTERT SV40T cells efficiently triggered the formation of filopodia (in 54.5 ± 4.1% of the cells). In contrast, the mutation did not abolish triggering of the formation of broad stress fibers, which was seen in 70.3 ± 12.3% of the cells (Figure 2A–C).

### 3.3. RhoGDI Binding Is Not Necessary for Cdc42-Dependent Actin Reorganization

The insert domain of Cdc42 has also been implicated in binding to RhoGDI [27,28]. This protein binds geranyl-geranylated classical Rho GTPases and sequesters them in the GDP-loaded conformation [9]. Arginine 66 of Cdc42 has been shown to be important for the interaction with RhoGDI [29]. To this end, this arginine 66 was mutated, with the alanine mutant Cdc42 created in the wt, Q61L, and F28L backgrounds. However, introduction of the R66A mutation only marginally affected the cellular responses of Cdc42/wt, Cdc42/Q61L, and Cdc42/F28L (Figure 3A–C, Appendix A), meaning that it is less likely that the RhoGDI interaction has a direct role in determining whether Cdc42 expression can trigger the formation of filopodia or broad lamellipodia, or can trigger the formation of broad stress fibers, or shows stress fiber loss.

### 3.4. Membrane Targeting Is Necessary for Cdc42-Dependent Actin Reorganization

Another factor that is essential for the function of Rho GTPases is the posttranslational geranyl-geranylation in their C-terminal CAAX boxes. The classical Rho GTPases undergo a series of modifications before they become fully functional. The initial modification occurs by the covalent addition of a 20-carbon geranylgeranyl to the cysteine residue in the CAAX motif, which is catalyzed by the enzyme known as geranylgeranyltransferase-1 (GGT-1) in the ER [30]. After this step, the AAX tripeptide is removed by the proteolytic activity of Ras and a-factor converting enzyme (Rce1), which is followed by carboxylmethylation of the prenylated cysteine by isoprenylcysteine carboxyl methyltransferase (Icmt). To inhibit the geranyl-geranylation of Cdc42, the cysteine in the CAAX box was mutated to a serine in Cdc42/Q61L and Cdc42/F28L. The resulting SAAX mutants were transfected into the BJ/hTERTSV40T cells as before. Interestingly, this modification abrogated the cellular responses elicited by intact Cdc42/Q61L and Cdc42/F28L (Figure 3A–C, Appendix A). These cells appeared to have normal cell edges and normal stress fibers, but they were notably elongated and had an increased perimeter (Figure 3D,E). The cell area occupied by the Cdc42/F28LSAAX mutant was smaller than the area occupied by the Cdc42/F28L mutant with the intact CAAX box but mutating the CAAX box in the Cdc42/Q61L background did not affect the cell area (Figure 3F). This indicated that these CAAX-box mutants can influence cell morphology, even though they do not trigger the formation of broad lamellipodia or filopodia (Figure 3A).

Whether an inhibitor of GGT-1 (GGTI298) also abolished the cellular responses of Cdc42/wt, Cdc42/Q61L, and Cdc42/F28L was then also tested, since inhibiting GGT-1 could be reasoned to give similar cellular effects as mutating the CAAX box. To this end, the transfected cells were treated with 10 μM GGTI298 for 18 h. This caused a loss of adhesion and stress fibers in the non-transformed cells. However, despite the loss of filamentous actin, it was possible to conclude that treatment with GGTI298 did not influence the basic cellular responses triggered by these Cdc42 variants (Appendix A). This suggests that Cdc42 can be posttranslationally modified by other mechanisms. To determine whether Cdc42 can be farnesylated or palmitoylated, Cdc42/wt, Cdc42/Q61L, and Cdc42/F28L were treated with an inhibitor of farnesyl transferase (FTI277) or with the protein palmitoylation inhibitor 2-bromopalmitate (2-BP). These treatments with 10 μM FTI277 or 100 μM 2-BP for 18 h did not visibly interfere with the cellular responses elicited by the Cdc42 variants (Appendix A). This indicated that mutation of the CAAX box constitutes a more definitive method to inactivate the posttranslational modifications of Cdc42.

### 3.5. Rac1 Mutants with Elevated Intrinsic GDP/GTP Exchange Activities Induce Filopodia

The fast-cycling Cdc42 variants triggered the formation of filopodia, and, therefore, it was of interest to determine whether the fast-cycling mutants of Rac1 are also involved in filopodia formation. To this end, the BJ/hTERTSV40T cells were transfected with Rac1wt, Rac1/Q61L, Rac1/F28L, and Rac1/T17N expected (Figure 4A–C, Appendix A). The GTPase-deficient mutant Rac1/Q61L effectively induced the formation of broad lamellipodia and broad stress fibers (in 74.5 ± 3.3% and 76.1 ± 3.6% of the cells), which was as expected (Figure 4A–C). Interestingly, neither Rac1/wt nor Rac1/F28L induced lamellipodia formation. In contrast, 58.4 ± 13.5% of the Rac1/wt-expressing cells and 49.7 ± 17.5% of the Rac1/F28L-expressing cells developed filopodia in transfected cells (Figure 4A,B). This response was accompanied by a dramatic loss of stress fibers in both the Rac1/wt- and Rac1/F28L-expressing cells (in 69.8 ± 2.1% and 68.4 ± 12.2% of the cells) (Figure 4A,C). A similar response was seen in the PAE/PDGFRβ cells (Appendix A). This demonstrated that active GDP/GTP exchange activity is a prerequisite for Rho GTPases to induce filopodia formation.

Rac1B is a cancer-associated fast-cycling splice variant of Rac1 [20,21]. Expression of Rac1B in the BJ/hTERT SV40T cells promoted filopodia formation (in 50.4 ± 7.6% of the cells). However, in contrast to Rac1/wt and Rac1/F28L, Rac1B triggered stress fiber loss in only 9.6 ± 2.6% of the cells (Figure 4A,C). A Q61L mutation in the Rac1B background showed a reduced formation of broad lamellipodia (in 5.6 ± 4.6% of the cells, compared to 74.5 ± 3.3% in the Rac1/Q61L-expressing cells) and broad stress fibers (Figure 4B,C, Appendix A). Finally, triggering of formation of filopodia and dissolution of stress fibers were tested for another cancer-associated Rac1 mutant, Rac1/P29S. The recurrent point mutation Rac1/P29S has been implicated in melanomas and has been shown to have significantly increased intrinsic GDP/GTP exchange activity [31]. Remarkably, when expressed in the BJ/hTERT SV40T cells, Rac1/P29S resulted in filopodia formation in 68.2 ± 7.6% of the cell, accompanied by stress fiber loss in 42.8 ± 10.9% of the cells (Figure 4A–C). Similar to Cdc42, BJ/hTERT SV40T cells expressing Rac1/wt, Rac1/Q61L, and Rac1/F28L were treated with 10 μM GGTI298, 10 μM FTI277, or 100 μM 2-BP for 18 h. These treatments did not visibly affect the cytoskeletal reorganization induced by these respective Rac1 variants (Appendix A).

### 3.6. The Involvement of Formins and Arp2/3 in Cdc42- and Rac1-Induced Actin Reorganization

Elevated intrinsic exchange activity appears to be a strong determinant for Cdc42 and Rac1 triggering of the formation of filopodia, and a GTPase-defective condition appears to be decisive for the formation of broad lamellipodia. What other factors might determine the fate of this cytoskeletal reorganization? To shed light on this question, the BJ/hTERT SV40T cells were transfected with the following constructs: Cdc42/Q61L, Cdc42/F28L, Rac1/Q61L, Rac1/F28L, and Rac1/P29S. Thereafter, these cells were treated with inhibitors of Src kinases (2 μM SU6656), PI3 kinase (10 μM LY294002), Rho kinase (10 μM Y27632), Rac1 (30 μM NSC23766), and Cdc42 (10 μM ML-141). SU6656 and LY294002 did not affect the phenotype of the Cdc42 and Rac1 mutants (Appendix A). Importantly, treatment of the cells expressing Cdc42/Q61L and Rac1/Q61L with the ROCK inhibitor Y27632 abrogated the formation of broad stress fibers, which indicated that the formation of broad stress fibers is dependent on Rho kinases (Appendix A). Interestingly, Y27632 treatment of Cdc42/Q61L transfected cells resulted in increased formation of filopodia (12.5 ± 3.5% to 43.0 ± 4.2% of the transfected cells), and decreased formation of broad lamellipodia (from 59 ± 1.4% down to 11 ± 0%) (Appendix A). In contrast, broad lamellipodia induced by Rac1/Q61L were not sensitive to Y27632 treatment (Appendix A). Moreover, treatment of Cdc42/Q61L-expressing cells with the Rac1 inhibitor NSC23766 also shifted the balance from broad lamellipodia (from 59 ± 1.4% in control cells to 19.9 ± 2.7%) to filopodia (from 12.5 ± 3.5% in the control cells up to 44.2 ± 1.8%). This is in agreement with the notion that Cdc42-dependent lamellipodia formation is dependent on Rac1 (Appendix A) [3]. In contrast, treatment with the Cdc42 inhibitor ML-141 did not interfere with any of the Rac1-induced cellular responses, indicating that Cdc42 is not acting downstream of Rac1 in filopodia formation (Appendix A).

As described above, mutations in the effector binding sites of Cdc42 can switch the cytoskeletal response from broad lamellipodia to filopodia (notably for Cdc42/Q61LF37A, Cdc42/Q61T35A, Cdc42/Q61L). To determine which categories of effectors are involved in these responses, the BJ/hTERT SV40T cells were transfected with Cdc42/Q61L, Cdc42/F28L, Rac1/Q61L, Rac1/F28L, or Rac1/P29S, and then they were treated with 30 μM SMIFH2, an inhibitor of formins, or 100 μM CK-666, an inhibitor of Arp2/3, for 18 h [32,33]. This inhibition of Arp2/3 with CK-666 effectively inhibited the formation of broad lamellipodia induced by the Cdc42/Q61L (down from 59 ± 1.4% in control cells to 45 ± 0.7%) and Rac1/Q61L mutants (down from 68 ± 7.1% to 16 ± 1.4%), and instead, the cells formed numerous filopodia (Figure 5A–C). However, their effects on broad stress fibers formation were not affected (Figure 5A,B,D). In contrast, inhibition of formins did not interfere with the formation of broad lamellipodia and stress fibers by expression of Cdc42/Q61L or Rac1/Q61L (Figure 5A–C).

Interestingly, formin inhibition by SMIFH2 treatment in cells expressing Rac1/F28L or Rac1/P29S interfered with their ability for filopodia formation and, instead, an increased formation of broad lamellipodia was noticed (Figure 5B,C). SMIFH2 treatment of Rac1/F28L-expressing cells reduced the number of cells with filopodia from 62 ± 2.8% down to 17.9 ± 2.7% and in Rac1/P29S, the number of cells with filopodia decreased from 62 ± 4.2% to 12 ± 4.2%. A similar, albeit weaker, tendency was also detected with formin inhibition of Cdc42/F28L-expressing cells, from 53 ± 4.2% down to 30.4 ± 6.2% (Figure 5A,C). Moreover, formin inhibition was associated with a shift from stress fiber dissolution to the formation of broad stress fibers in cells expressing the fast-cycling Rac1/F28L (from 76.5 ± 4.9% down to 16.9 ± 4.1%) and Rac1/P29S (from 45.5 ± 0.7% down to 11.5 ± 4.5%), and in cells expressing Cdc42/F28L (from 55.5 ± 9.2% down to 26.4 ± 2.3%) (Figure 5A,B,D). Finally, Arp2/3 inhibition of the cells expressing the fast-cycling mutants of Cdc42 and Rac1 did not influence their ability to trigger filopodia formation or stress fiber dissolution (Figure 5 A,B,D).

## 4. Discussion

In the present study, the cellular effects produced by GTPase-deficient and fast-cycling mutants of Cdc42 and Rac1 were compared. The GTPase-deficient mutants of Cdc42 and Rac1 were found to trigger predominantly the formation of broad lamellipodia and the assembly of broad stress fibers. In contrast, the fast-cycling mutants triggered the formation of filopodia, accompanied by the dissolution of stress fibers. One possible key to the different behaviors of these two categories of mutant Rho GTPases resides in their intrinsic kinetic properties. GTPases-deficient variants, such as the Q61L and G12V mutants of Cdc42 and Rac1, are locked in the activated GTP-bound conformation, whereas the wild-type and fast-cycling mutants, such as F28L and P29S, actively cycle between GDP-bound and GTP-bound conformations. A plausible interpretation is therefore that the GDP/GTP exchange activity is an important determinant for the ability of Cdc42 and Rac1 to trigger the formation of filopodia. This is in line with the finding that the atypical fast-cycling atypical Rho GTPases RhoU, RhoD, and RhoF are involved in the formation of filopodia [11]. The fact that an elevated intrinsic GDP/GTP is associated with filopodia formation does not exclude the possibility that accessory proteins can be involved in the regulation of the exchange activity of the fast-cycling mutants. For instance, Cdc42/F28L was found to have an intact GTPase activity, which could be further stimulated by Cdc42-GAP, suggesting that it is not the intrinsic exchange activity alone that determines the outcome of Cdc42/F28L [15]. Interfering with the RhoGDI-binding of Cdc42 did not influence the cellular responses elicited by the GTPase-deficient and fast-cycling variants of Cdc42. However, when the serine residue in the CAAX box of Cdc42 was mutated in Cdc42/Q61L and Cdc42/F28L, to abrogate post-translational geranyl-geranylation, the resulting Cdc42/Q61LSAAX and Cdc42/F28LSAAX mutants did not localize to the cell periphery, and they did not induce broad lamellipodia or filopodia.

The seminal studies by Alan Hall and his colleagues resulted in the current paradigm stating that the concerted activation of the three Rho GTPases Cdc42, Rac1, and RhoA are responsible for the dynamic reorganization of the actin filament system [23]. In addition to the concept that states that Rac1 regulates the formation of lamellipodia, Cdc42 the formation of filopodia, and RhoA the assembly of stress fibers, there is a hierarchical relationship between these three Rho GTPases, whereby Cdc42 is upstream of Rac1, which, in turn, acts upstream of RhoA [3]. The observations presented in the current study do not necessarily contradict this concept but argue that the roles of these three atypical Rho GTPases are more intricate than indicated in the current paradigm, since fast-cycling mutants of Rac1 can trigger the formation of filopodia independently of Cdc42. There is, in fact, conflicting data regarding the absolute requirement of Cdc42 for the formation of filopodia. Studies in the groups of Brakebusch and Rottner have shown that mouse fibroblastoid cells lacking Cdc42 can still form filopodia and undergo normal directed cell migration [34]. In contrast, the group of Yi Zheng reported a significant reduction of filopodia formation and directed cell migration in mouse embryonic fibroblasts (MEFs) that lacked Cdc42 [35]. The data presented in the current study is compatible with a model of Cdc42 preceding Rac1 in the formation of broad lamellipodia, since treatment with NSC23766 resulted in decreased lamellipodia formation in cells expressing GTPase-deficient Cdc42. NSC23766 blocks the interface between Rac1 and the RhoGEFs Tiam-1 and Trio, and possible other RhoGEFs, which suggests that Cdc42 activation of Rac1 could involve Tiam-1 or Trio [36]. The ML-141 inhibitor is thought to act directly on Cdc42 and not by interfering with the RhoGEF:Cdc42 interface [37]. Inhibition of Cdc42 with ML-141 treatment, did not visibly interfere with the Rac1-dependent cellular responses, suggesting that Cdc42 is not acting downstream of Rac1 in the formation of filopodia induced by the fast-cycling mutants of Rac1. Treatment of Rac1/Q61L- or Cdc42/Q61L-expressing cells with the ROCK inhibitor Y27632 abrogated the formation of broad stress fibers. Interestingly, Y27632 also interfered with the Cdc42/Q61L-induced lamellipodia formation and shifted the balance towards filopodia formation in cells expressing this Cdc42 mutant. This suggests that the Cdc42-dependent formation of broad lamellipodia is both dependent on Rac1 and ROCK. In contrast, Y27632 did not inhibit broad lamellipodia induced by Rac1/Q61L expression.

As shown in the current study, GTPase deficiency is an important determinant for the formation of broad lamellipodia, but which categories of effectors are needed to mediate this effect? Analysis of Cdc42/Q61L with mutations to the effector loop showed that Cdc42/Q61LF37A and Cdc42/Q61LT35A no longer induced the formation of broad lamellipodia. However, Cdc42/Q61LY40C still triggered the formation of broad lamellipodia. Mutations in tyrosine 40 abolished the interactions with CRIB domain-containing effectors, such as WASP and PAK, which suggested that these proteins are not involved in lamellipodia formation downstream of GTPase-deficient Cdc42 [26,38]. Mutation of threonine 35 has been shown to have a more general negative influence on effector interactions in the context of both Cdc42 and Ras, whereas mutation of phenylalanine 37 only results in small decreases in the affinities for PAK and WASP [38,39,40]. The T35A mutation in the Cdc42/Q61L background abrogates both the lamellipodia response and formation of broad stress fibers, whereas the Cdc42/Q61LF37A mutant can still trigger the formation of broad lamellipodia. This latter effect is also seen in cells that express the Cdc42/Q61LΔins mutant. Interestingly, treatment of Cdc42/Q61L- or Rac1/Q61L-expressing cells with the Arp2/3 inhibitor CK-666 shifted the cellular responses from broad lamellipodia to filopodia formation, which suggests that effectors, which act via Arp2/3-mediated actin reorganization, are involved in the response. However, possibly not the WASP family of actin of Arp2/3-binding proteins, since Cdc42/Q61LY40C, which does not bind CRIB domains, could still induce broad lamellipodia. Instead, the WAVE family of proteins are possible candidates for the formation of broad lamellipodia downstream of Cdc42 and Rac1 [41]. Arp2/3 inhibition did not interfere with the activity of the fast-cycling Cdc42 or Rac1 mutants. Importantly, the treatment of the fast-cycling mutants of Cdc42 and Rac1 with SMIFH2, the inhibitor of formin-dependent actin reorganization, interfered with filopodia formation and stress fiber dissolution, indicating a critical role for formins in these responses [42].

In summary, the effects of the classical Rho GTPases Cdc42 and Rac1 on the dynamic reorganization of the actin filament system are, to a large extent, defined by their catalytic activities. GTPase-deficient mutants can trigger the formation of broad lamellipodia and stress fibers, and the fast-cycling mutations can trigger filopodia formation and stress fiber dissolution. The filopodia response requires the activity of the formin family of actin nucleation promotors. In contrast, the effects of GTPase-deficient Cdc42 and Rac1 are mediated through Arp2/3-dependent actin nucleation, although probably not through the WASP family of Arp2/3-binding proteins.

## Figures and Tables

**Figure 1 cells-08-00759-f001:**
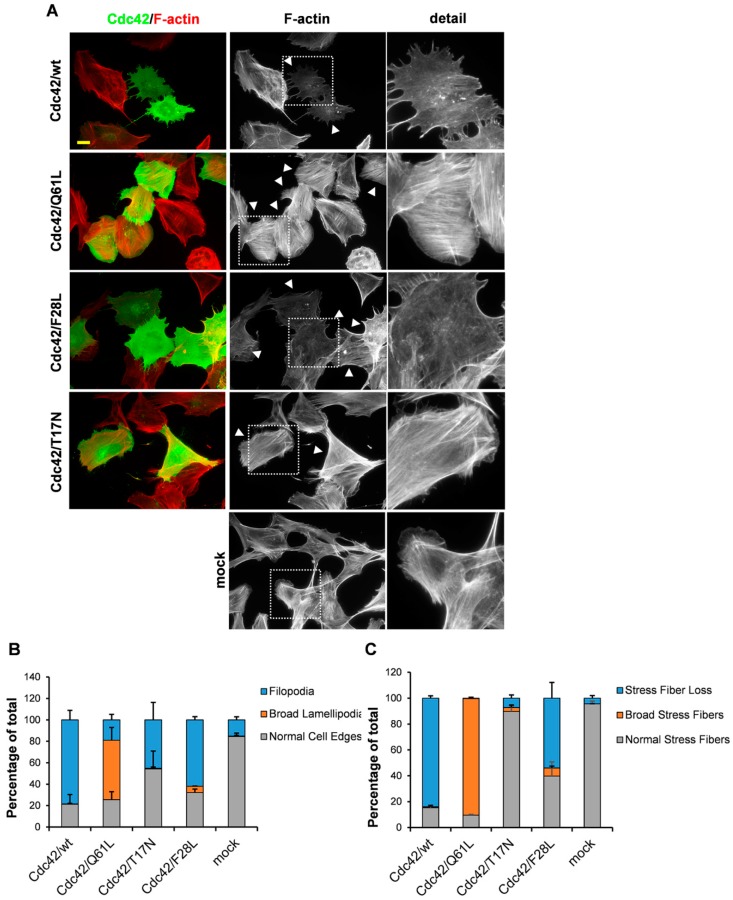
Cdc42 effects on actin dynamics. (**A**) Myc-tagged wt and mutant Cdc42 were exogenously expressed in BJ/hTERTSV40T cells. Myc-tagged proteins were detected with a rabbit anti-Myc antibody, followed by an Alexa Fluor 488-conjugated donkey anti-rabbit antibody. Filamentous actin was visualized using TRITC-conjugated phalloidin. Arrow-heads mark transfected cells. The boxed areas are enlarged at the right-hand-side of the corresponding image. Scale bar, 20 µm. (**B**,**C**) Quantification of formation of filopodia and broad lamellipodia (**B**), and of actin filament organization (**C**). At least 100 transfected cells were scored for each aspect (as indicated) from three independent experiments. Data are means ± standard deviation.

**Figure 2 cells-08-00759-f002:**
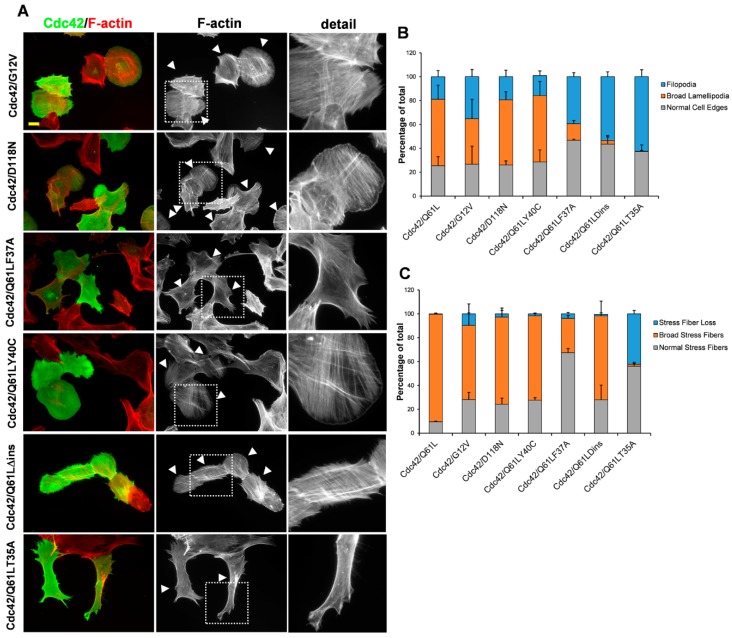
Cdc42 effects on actin dynamics (continued). (**A**) Myc-Cdc42/G12V, Myc-Cdc42/D118N, Myc-Cdc42/Q61LF37A, Myc-Cdc42/Q61L/Y40C, Myc-Cdc42/Q61LΔins, and Myc-Cdc42/Q61LT35A were exogenously expressed in BJ/hTERT SV40T cells. Myc-tagged Cdc42 was detected with a rabbit anti-Myc antibody, followed by an Alexa Fluor 488-conjugated donkey anti-rabbit antibody. Filamentous actin was visualized using TRITC-conjugated phalloidin. Arrow-heads mark transfected cells. The boxed areas are enlarged at the right-hand-side of the corresponding image. Scale bar, 20 µm. (**B**,**C**) Quantification of formation of filopodia and broad lamellipodia (**B**), and of actin filament organization (**C**). At least 100 transfected cells were scored for each aspect (as indicated) from three independent experiments. Data are means ± standard deviation.

**Figure 3 cells-08-00759-f003:**
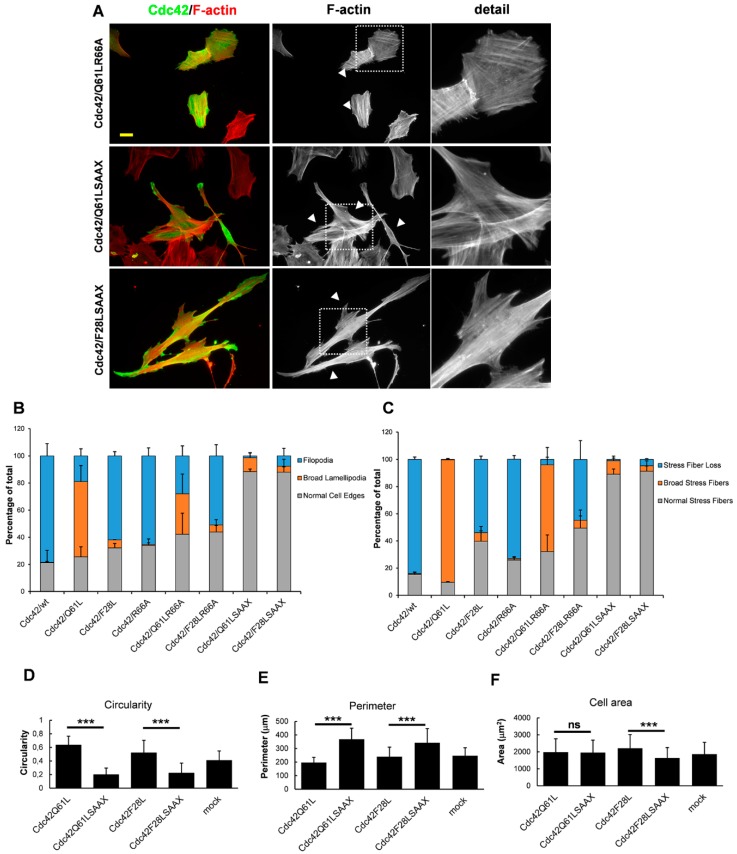
The involvement of CAAX box and RhoGDI in the Cdc42-induced actin reorganization. (**A**) Myc-tagged Cdc42/Q61LR66A and Cdc42 variants with a mutation in the CAAX-box (Cdc42/Q61LSAAX, Cdc42/F28LSAAX) were exogenously expressed in BJ/hTERT SV40T cells. Myc-tagged proteins were detected with a rabbit anti-Myc antibody, followed by an Alexa Fluor 488-conjugated donkey anti-rabbit antibody. Filamentous actin was visualized using TRITC-conjugated phalloidin. Arrow-heads mark transfected cells. The boxed areas are enlarged at the right-hand-side of the corresponding image. Scale bar, 20 µm. (**B**,**C**) Quantification of formation of filopodia and broad lamellipodia (**B**), and of actin filament organization (**C**). At least 100 transfected cells were scored for each aspect (as indicated) from three independent experiments. Data are means ±standard deviation. Differences in cell shape between Cdc42/Q61L and Cdc42/Q61LSAAX, and Cdc42/F28L and Cdc42/F28LSAAX were determined using ImageJ as described in the Experimental section. (**D**), circularity, (**E**) perimeter and (**F**) cell area. Unpaired two-way Student’s *t*-tests with unequal variance were performed to calculate statistical significance. Histogram shows mean values and error bars standard deviation. *** = *p* < 0.001, ns = non-significant.

**Figure 4 cells-08-00759-f004:**
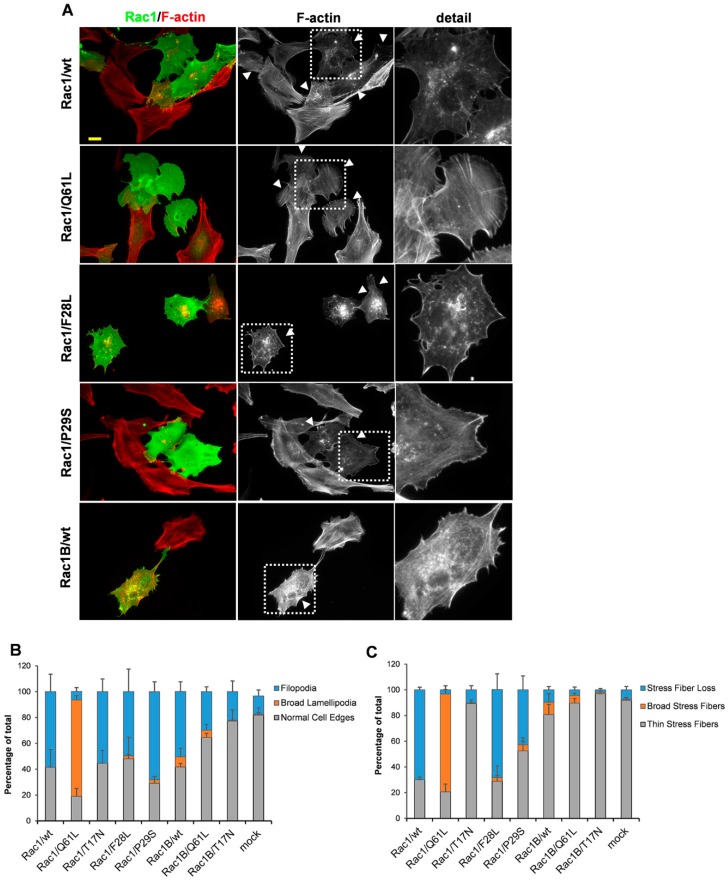
Rac1 effects on actin dynamics. (**A**) Myc-tagged wt and mutant Rac1 were exogenously expressed in BJ/hTERTSV40T cells. Myc-tagged proteins were detected with a rabbit anti-Myc antibody followed by an Alexa Fluor 488-conjugated donkey anti-rabbit antibody. Filamentous actin was visualized using TRITC-conjugated phalloidin. Arrow-heads mark transfected cells. The boxed areas are enlarged at the right-hand-side of the corresponding image. Scale bar, 20 µm. (**B**,**C**) Quantification of formation of filopodia and broad lamellipodia (**B**), and of actin filament organization (**C**). At least 100 transfected cells were scored for each phenotype (as indicated) from three independent experiments. Data are means ±standard deviation.

**Figure 5 cells-08-00759-f005:**
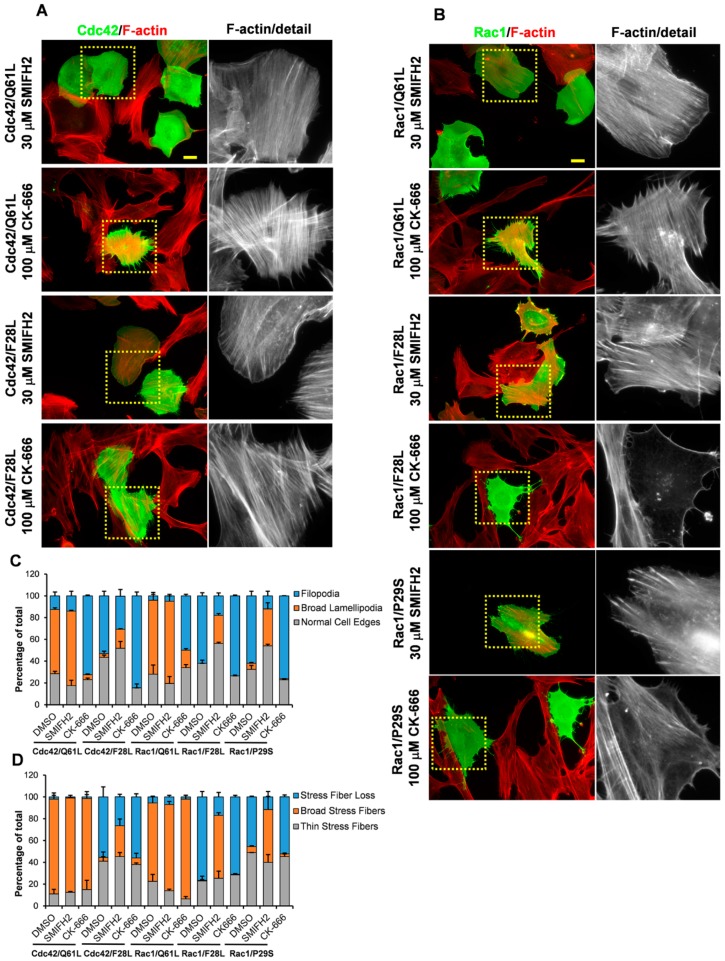
The involvement of formins and Arp2/3 in Cdc42- and Rac1-induced actin reorganization. (**A**,**B**) Myc-tagged Cdc42/Q61L or Myc-tagged Cdc42/F28L (**A**), and Myc-tagged Rac1/Q61L, Myc-tagged Rac1/F28L, or Myc-tagged Rac1/P29S (**B**), were exogenously expressed in BJ/hTERT SV40T cells. Six hours after transfection, the cells were treated with 30 μM SMIFH2 or 100 μM CK-666 for 18 h, and then fixed. Myc-tagged protein was detected with a rabbit anti-Myc antibody followed by an Alexa Fluor 488-conjugated donkey anti-rabbit antibody. Filamentous actin was visualized using TRITC-conjugated phalloidin. Arrow-heads mark transfected cells. The boxed areas are enlarged at the right-hand-side of the corresponding image. Scale bar, 20 µm. (**C**,**D**) Quantification of formation of filopodia and broad lamellipodia (**C**), and of actin filament organization (**D**). At least 200 transfected cells were scored for each phenotype (as indicated).

**Table 1 cells-08-00759-t001:** **A.** Cdc42 and Rac1 variants used in the study. **B**. Inhibitors used in the study.

**A. Cdc42 and Rac1 variants used in the study**		
**Small GTPase**	**Mutant**	**Phenotype**
**Cdc42**	Q61L	GTPase defective
	G12V	GTPase defective
	T17N	Dominant negative (nucleotide binding-defective)
	F28L	Fast-cycling (increased GDP/GTP exchange)
	D118N	Elevated GDP/GTP exchange
	R66A	RhoGDI binding-defective
	T35A	Effector loop mutant
	F37A	Effector loop mutant
	Y40C	Effector loop mutant
	Δins	Insert domain mutant
	SAAX	CAAX box mutant
**Rac1**	Q61L	GTPase defective
	T17N	Dominant negative (nucleotide binding- defective)
	F28L	Fast-cycling (increased GDP/GTP exchange)
	P29S	Fast-cycling (increased GDP/GTP exchange). Cancer mutation
	Rac1B	Fast-cycling (increased GDP/GTP exchange). Cancer mutation
**B. Inhibitors used in the study**		
**Inhibitor**	**Concentration Used**	**Targeted Pathway**
**GGTI298**	10 μM	Inhibitor of geranylgeranylation
**FFT277**	10 μM	Inhibitor of farnesylation
**2-bromopalmitate (2-BP)**	100 μM	Inhibitor of palmitoylation
**SU6656**	2 μM	Inhibitor of Src family kinases
**LY294002**	10 μM	Inhibitor of PI3 kinases
**Y27632**	10 μM	Inhibitor of Rho kinase (ROCK)
**NSC23766**	30 μM	Inhibitor of Rac
**ML-141**	10 μM	Inhibitor of Cdc42
**SMIFH2**	30 μM	Inhibitor of formins
**CK-666**	100 μM	Inhibitor of Arp2/3

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
