# Peer review of "The Intrinsic GDP/GTP Exchange Activities of Cdc42 and Rac1 Are Critical Determinants for Their Specific Effects on Mobilization of the Actin Filament System"

_cells, 2019, doi:10.3390/cells8070759_

Reviewer 1 Report

In this study, the author transiently and ectopically expressed various mutants of Cdc42, Rac1, and Rac1B, and examined their effects on lamellipodia, filopodia, and stress fiber formation. In contrast to the well-known phenomenon that the GTPase-deficient mutant of Cdc42 and Rac1 triggers formation of broad lamellipodia and stress fibers, the author found that the fast-cycling mutant triggered filopodia formation and stress fiber dissolution, leading to the conclusion that the intrinsic GDP/GTP exchange activities of Cdc42 and Rac1 are critical determinants for their specific effects on F-actin structures.

However, when these two classes of mutants were compared in this study, their expression levels in cells did not appear strictly controlled. For example, in Figure 3D, the fast-cycling Cdc42/F28L mutant was expressed significantly lower than the GTPase-deficient Cdc42/Q61L mutant, implying intrinsic instability of the fast-cycling mutant protein in vivo. To eliminate effects caused by unequal expression levels in transient transfection, all microscopic examinations in this study needed extremely careful selection of cells showing similar expression levels of the mutant proteins that were compared, but such selection did not appear carried out. Therefore, another possibility can remain feasible that different effects on F-actin structures by the GTPase-deficient mutant and the fast-cycling mutant are simply due to unequal cellular abundance of these two classes of mutants. Hence, the presented data set does not well substantiate the authors’ conclusion.

Other points

The fast-cycling mutant but not the GTPase-deficient mutant can be inactivated by RhoGAPs. Therefore, even when these two classes of mutants are comparably expressed in cells, the fast-cycling mutant may have different physiological impact from that of the GTPase-deficient mutant, due to lower abundance of the GTP-bound active form of the fast-cycling mutant in cells. This could be another explanation for the different effects on F-actin structures between the GTPase-deficient mutant and the fast-cycling mutant.

In section 3.6, Cdc42/Q61L triggered filopodia formation in the presence of the ROCK inhibitor Y27632. Therefore, unless the ROCK inhibitor restores GDP/GTP exchange activity in Cdc42/Q61L, the title of section 3.1 “Cdc42-induced filopodia formation requires intact GDP/GTP exchange activity” does not appear correct.

Charts and their associated labels in all main figures are too small and far from readable.

Author Response

 Reviewer #1

In this study, the author transiently and ectopically expressed various mutants of Cdc42, Rac1, and Rac1B, and examined their effects on lamellipodia, filopodia, and stress fiber formation. In contrast to the well-known phenomenon that the GTPase-deficient mutant of Cdc42 and Rac1 triggers formation of broad lamellipodia and stress fibers, the author found that the fast-cycling mutant triggered filopodia formation and stress fiber dissolution, leading to the conclusion that the intrinsic GDP/GTP exchange activities of Cdc42 and Rac1 are critical determinants for their specific effects on F-actin structures.

However, when these two classes of mutants were compared in this study, their expression levels in cells did not appear strictly controlled. For example, in Figure 3D, the fast-cycling Cdc42/F28L mutant was expressed significantly lower than the GTPase-deficient Cdc42/Q61L mutant, implying intrinsic instability of the fast-cycling mutant protein in vivo.

The Western blot data has been removed in the revised article, since it did not add to the data presented in this study (this is discussed in my reply to reviewer #3). It is worth noticing that the Western blot data was produced in another cell system (HEK293T cells) using a different transfection protocol (Lipofectamine). When analyzing the transfected BJ/hTERTSV40T cells I can notice differences in the number of cells that have been transfected between the different DNA constructs but, importantly, the expression levels in the cells that actually have been transfected are relatively equal.

To eliminate effects caused by unequal expression levels in transient transfection, all microscopic examinations in this study needed extremely careful selection of cells showing similar expression levels of the mutant proteins that were compared, but such selection did not appear carried out.

I am fully aware of this problem and have carefully selected cells with equal expression levels, i.e. equal intensity of cell staining. Cells with low and high expression were excluded from the analysis. Information about the criteria for the analysis is included in the revised supplementary figures S1 and in the new figure S2. See also my comments to Reviewer #3.

Therefore, another possibility can remain feasible that different effects on F-actin structures by the GTPase-deficient mutant and the fast-cycling mutant are simply due to unequal cellular abundance of these two classes of mutants. Hence, the presented data set does not well substantiate the authors’ conclusion.

I am confident that I have studied true cellular effects. I do not see major differences in the number of transfected cells or in expression levels between the two categories. I have been careful in selecting cells with similar expression profiles in order to minimize handling artifacts.

Other points

The fast-cycling mutant but not the GTPase-deficient mutant can be inactivated by RhoGAPs. Therefore, even when these two classes of mutants are comparably expressed in cells, the fast-cycling mutant may have different physiological impact from that of the GTPase-deficient mutant, due to lower abundance of the GTP-bound active form of the fast-cycling mutant in cells. This could be another explanation for the different effects on F-actin structures between the GTPase-deficient mutant and the fast-cycling mutant.

It is true that at least Cdc42/F28L has an intact GTPase activity, which can be stimulated by Cdc42-GAP. I have mentioned this possibility in the revised discussion.

In section 3.6, Cdc42/Q61L triggered filopodia formation in the presence of the ROCK inhibitor Y27632. Therefore, unless the ROCK inhibitor restores GDP/GTP exchange activity in Cdc42/Q61L, the title of section 3.1 “Cdc42-induced filopodia formation requires intact GDP/GTP exchange activity” does not appear correct.

I have toned down the statement in order to avoid claiming that and intact GDP/GTP exchange activity is an absolute requirement. However, the GDP/GTP exchange activity is one factor that determines the biological outcome. The revised title reads “The GDP/GTP exchange activity is the basis for Cdc42-induced filopodia formation”.

Charts and their associated labels in all main figures are too small and far from readable.

Yes, I agree with the reviewer. I have increased the size of the embedded images and text when possible and I have enclosed high-resolution figures, which I hope will increase the visibility of the figures. 

Reviewer 2 Report

In this manuscript entitled: “The intrinsic GDP/GTP exchange activities of Cdc42

and Rac1 are critical determinants for their specific effects on mobilization of the actin filament system by  Pontus Aspenström, the author states that this study demonstrates that the effects of the classical Rho GTPases Cdc42 and Rac1 on the dynamic reorganization of the actin filament system are defined by their catalytic activities.

MAJOR CONCERNS

Although the topic of the MS is quite interesting, there are some major concerns that arise, including:

1.      Description of the images from the immunocytochemistry (not immunohistochemistry as stated) should be more detailed and more information should be given in the Results section as well as in the legends of the respective figures. For example, what do the arrows that appear in the images point at? What do the squares drawn specify?

2.      All the percentages reported lack mentioning their respective standard deviation. Thus, the author should not just report a number but also the ±….  percentage!

3.      Another type of graph might be more informative and easier to comprehend than the one chosen and presented by the author.

4.      Finally, although sound, the scope of the study and its potential contribution to the specific scientific field are lacking and should become more evident in the Discussion. This should not be just a descriptive study but also refer to the ways the findings support, enhance or improve our current knowledge in the field. A perspective should be presented.

Author Response

Reviewer #2

In this manuscript entitled: “The intrinsic GDP/GTP exchange activities of Cdc42 and Rac1 are critical determinants for their specific effects on mobilization of the actin filament system by  Pontus Aspenström, the author states that this study demonstrates that the effects of the classical Rho GTPases Cdc42 and Rac1 on the dynamic reorganization of the actin filament system are defined by their catalytic activities.

MAJOR CONCERNS

Although the topic of the MS is quite interesting, there are some major concerns that arise, including:

1.        Description of the images from the immunocytochemistry (not immunohistochemistry as stated) should be more detailed and more information should be given in the Results section as well as in the legends of the respective figures. For example, what do the arrows that appear in the images point at? What do the squares drawn specify?

I have corrected the error and included the requested information in the revised figure legends.

All the percentages reported lack mentioning their respective standard deviation. Thus, the author should not just report a number but also the ±….  percentage!

I agree of course. This has been corrected in the revised manuscript.

2.        Another type of graph might be more informative and easier to comprehend than the one chosen and presented by the author.

I have tested a number of differ graphs but they tend to become rather extensive and difficult to compare. I think that the current design is the best I could come up with, provided it is printed at sufficiently high resolution, something I need to ensure with help from the production department.  I noticed that Klemens Rottner had used a similar design of graphs in a recent paper (Steffen et al., JCS, 126, 4572) and I found this to be an elegant way to display complex datasets. If the reviewer still think this is a problem, I am off course happy if the reviewer could suggest an alternative design.

3.        Finally, although sound, the scope of the study and its potential contribution to the specific scientific field are lacking and should become more evident in the Discussion. This should not be just a descriptive study but also refer to the ways the findings support, enhance or improve our current knowledge in the field. A perspective should be presented.

It is tricky to navigate between the disparate suggestions from Reviewer #2 and Reviewer #3, who thinks that the discussion is too speculative. I have tried to balance the divergent requests from the reviewers in the revised discussion.

Reviewer 3 Report

Overall comments

In this manuscript, the author addresses the paradigm of the role of canonical Rho GTPases in regulation of cell morphology and migration through the reorganization of the actin cytoskeleton. The author focuses the study on Cdc42 and Rac1, and uses GTPase-deficient, fast-cycling, effector loop and CAXX box mutants, as well as different combinations of these mutations, and a plethora of inhibitors for these GTPase and some of their known effectors, to try clarifying their precise effects on cellular morphology and actin cytoskeleton architecture.

The premises of the study are valid and much relevant, since many of these mutants are used often as tools to modulate or mimic the activation of Rho GTPases in many contexts, and frequently their behavior fails to follow the established paradigm, generating conflicting results.

However, the collection of morphological data through the employed experimental approach raises significant concerns.

Major concerns:

1)      Were all the constructs confirmed by DNA sequencing? If not they must be sequenced to ascertain the correctness of the coding sequence.

2)      Overexpression of mutant GTPases cause enormous cellular stress that often compromises cell viability. One rough but effective way of monitoring this by microscopy is the staining of the cells’ nuclei with DAPI or Hoechst to check their integrity. Not doing so may lead to inconsistent and erroneous interpretation of the observed morphological effects.

3)      The morphologic evaluation criteria do not seem to account for the expression level of the GTPase mutants. As said above, overexpression can significantly compromise cellular integrity, so that a proper phenotypic evaluation of the effects of the different GTPase mutants requires one of the soring criteria to be the GTPase expression level, e.g,  low, intermedium and high expression, based on signal intensity (assuming that all images were acquired with equivalent settings). These criteria are needed to ascertain whether the observed phenotypic changes are associated to the signaling properties of the mutant GTPase or simply to the cellular havoc caused by the exceedingly high expression levels.

4)      For instance, some interesting observations can be made from the supplementary figures. In Fig S4, there is a low expressing cell to the bottom left corner of the two high Cdc42wt expressing cells, which, instead of the putative filopodia, presents beautifully formed lamellipodia, with clearly branched actin, and to the upper right corner there is another low expressing cell showing clearly defined membrane ruffles...

5)      Of note, the overall poor quality of the figures embedded in the manuscript make it difficult to assess morphological differences between high- and low- expressing cells.

6)      Still regarding the reported cytoskeleton-based morphological changes, it is unclear what the author defines as “broad stress fibers”. Again, Fig 1A has a very poor quality in the pdf, but from Fig S1 the indicated “broad stress fibers” could also represent a thinning of the actin bundles that have yet to fully depolarize. Also puzzling is why the structures defined as “broad lamellipodia” lack the branched actin architecture that usually characterizes these membrane protrusions. Both situations need to be a better addressed.

7)      Still concerning the morphological evaluation, besides including the above-mentioned additional considerations and criteria, the scoring of the phenotypical changes, being qualitative, should be made by more than one observer and, preferably, these should be blind to the identity of the mutant GTPase being analyzed.

Particular concerns:

1)      The F28L mutants do not appear to dissolve stress fibers in PAE/PDGFRb cells - rather the opposite. Also, the images of WT-Cdc42 do not appear to show a significant formation of filopodia in these cells, despite the values plotted in the graph.

2)      Considering the SD in Cdc42 12V expressing cells, it is unjustified to state that there is a clear shift towards flilopodia relative to Cdc42 61L.

3)      It may be due to the poor image quality in the pdf but no clear filopodia protrusions are apparent in the representative images for Cdc42Q61LT35A overexpression. The same applies for Cdc42Q61L delta ins.

4)      Immunoprecipitation:

a.       expression levels of the different constructs vary considerably and the amount of precipitated GTPases was not assessed in the IP WB.A reprobing of the membranes with an anti-myc Ab from a different species would provide this information allowing comparison of co-IPed GDI/mutant GTPase amounts.

b.      Comparing the IPs with the expression levels in the TCL WB, it appears that Q61L already has a much decreased affinity to GDI possibly, similar to that of Q61LR66A. Interestingly, with the possible exception of the insert region deletion, all other co-mutations appear to increase Q61L affinity for RhoGDI. This experiment needs to be properly clarified.

c.       Overall, the experiments of interfering with RhoGDI binding were inconclusive and their pertinence is unclear, since the binding of RhoGDI is impaired in GTP-bound GTPases.

5)      Overall, the experiments with prenylation inhibitors and CAXX mutants do not add to previous knowledge.

6)      Experiments with Rac1B strongly suggest that at least some of the structures induced by overexpressing fast-cycling mutants and identified as filopodia may actually be overexpression artifacts, since Rac1b has been shown to fail to signal to the cytoskeleton rearranging machinery – also seen here through the lack of response upon overexpression of the "classical" constitutively active Q61L mutant.

7)      The discussion is largely speculative and should be rewritten to focus on the relevant cytoskeletal changes observed after cellular integrity/viability and construct expression levels are accounted for.

Minor comments:

1)      The sentence “A large number of studies on Cdc42 and Rac1 have used the G12V and Q61L mutants of these GTPases. These mutations are known to block the hydrolysis activities of the small GTPases (14). It is therefore difficult to understand why they should be considered constitutively active.” is provocative but its reach has to be better explained, otherwise may seem paradoxical to the readers.

2)      “Importantly, the Rac1B variant turned out to have much higher intrinsic GDP/GTP exchange activity, and might therefore qualify as a fast-cycling mutant protein (20,21).” ; “Rac1B is a cancer-associated fast-cycling splice variant of Rac1 (20,21).”; Work from Peter Jordan’s lab on Rac1b higher GDP/GTP exchange activity  in vivo proceeds that of the cited reports (J Biol Chem. 2003, 278(50): 50442-8) and should be mentioned.

3)      Fig S3- no A panel; unclear which mutant is represented in the top images; cells expressing low and High GTPase levels show completely different phenotypes.

4)      Line 323 – “… from 89& to…”.

Author Response

Reviewer #3

In this manuscript, the author addresses the paradigm of the role of canonical Rho GTPases in regulation of cell morphology and migration through the reorganization of the actin cytoskeleton. The author focuses the study on Cdc42 and Rac1, and uses GTPase-deficient, fast-cycling, effector loop and CAXX box mutants, as well as different combinations of these mutations, and a plethora of inhibitors for these GTPase and some of their known effectors, to try clarifying their precise effects on cellular morphology and actin cytoskeleton architecture.

The premises of the study are valid and much relevant, since many of these mutants are used often as tools to modulate or mimic the activation of Rho GTPases in many contexts, and frequently their behavior fails to follow the established paradigm, generating conflicting results.

However, the collection of morphological data through the employed experimental approach raises significant concerns.

Major concerns:

1)        Were all the constructs confirmed by DNA sequencing? If not they must be sequenced to ascertain the correctness of the coding sequence.

All the constructs have all been confirmed by sequencing.

2)        Overexpression of mutant GTPases cause enormous cellular stress that often compromises cell viability. One rough but effective way of monitoring this by microscopy is the staining of the cells’ nuclei with DAPI or Hoechst to check their integrity. Not doing so may lead to inconsistent and erroneous interpretation of the observed morphological effects.

I’m fully aware of the matter and I always stain the cell nuclei with DAPI. I have omitted DAPI staining in the figures in order to make the figures less complicated. Only cells with intact nuclei have been selected for the quantifications. In the revised supplementary Fig.S2 DAPI staining is included in order to show the integrity of cell nuclei in the cells selected for the analysis.

3)        The morphologic evaluation criteria do not seem to account for the expression level of the GTPase mutants. As said above, overexpression can significantly compromise cellular integrity, so that a proper phenotypic evaluation of the effects of the different GTPase mutants requires one of the soring criteria to be the GTPase expression level, e.g., low, intermedium and high expression, based on signal intensity (assuming that all images were acquired with equivalent settings). These criteria are needed to ascertain whether the observed phenotypic changes are associated to the signaling properties of the mutant GTPase or simply to the cellular havoc caused by the exceedingly high expression levels.

This matter was also brought up by Reviewer #1 and I am certainly aware of this problem and the importance for defining strict criteria for which cells that should be selected for the analysis. I have incorporated more data about how to select cells in the revised manuscript, I have also included more examples of which cells should be selected or not selected in the supplementary Fig.S1 and S2.

4)        For instance, some interesting observations can be made from the supplementary figures. In Fig S4, there is a low expressing cell to the bottom left corner of the two high Cdc42wt expressing cells, which, instead of the putative filopodia, presents beautifully formed lamellipodia, with clearly branched actin, and to the upper right corner there is another low expressing cell showing clearly defined membrane ruffles...

As discussed above, it is possible to find differences in expression levs of the exogenous protein. This is why I have taken care to analyze cells with equal apparent expression level. The supplementary Fig.S2 gives examples of which cells to include and exclude in the analysis. It is also worth noticing that roughly 20% of the transfected cells have normal stress fibers and do not form filopodia. The two cells mentioned by the reviewer have not been included in the quantification, because the expression level is too low. However, in my view, they do not form “broad lamellipodia” and the stress fibers look normal so they do not look like the Cdc42Q61L-expressing cells.

5)        Of note, the overall poor quality of the figures embedded in the manuscript make it difficult to assess morphological differences between high- and low- expressing cells.

I regret that the figures have been reduced too much during the page editing. I have increased the size of the embedded text and images, when needed and I have submitted high-resolution images of the revised figures.

6)        Still regarding the reported cytoskeleton-based morphological changes, it is unclear what the author defines as “broad stress fibers”. Again, Fig 1A has a very poor quality in the pdf, but from Fig S1 the indicated “broad stress fibers” could also represent a thinning of the actin bundles that have yet to fully depolarize. Also puzzling is why the structures defined as “broad lamellipodia” lack the branched actin architecture that usually characterizes these membrane protrusions. Both situations need to be a better addressed.

I have included new images in the revised Fig.S1 to describe better the phenotypes.

7)        Still concerning the morphological evaluation, besides including the above-mentioned additional considerations and criteria, the scoring of the phenotypical changes, being qualitative, should be made by more than one observer and, preferably, these should be blind to the identity of the mutant GTPase being analyzed.

I agree with the reviewer that the ideal experimental design would to do the analysis in blind. Regrettably, I have no people in my lab, which means that I have to carry out all experiments myself (that is why I am the sole author of this article). Having this said, the phenotypes are rather robust. It is for instance immediately obvious which cells express Cdc42/Q61L and Rac1/Q61L.

Particular concerns:

1)        The F28L mutants do not appear to dissolve stress fibers in PAE/PDGFRb cells - rather the opposite. Also, the images of WT-Cdc42 do not appear to show a significant formation of filopodia in these cells, despite the values plotted in the graph.

The PAE/PDGFRbeta cells were not quantified for stress fiber loss or formation of broad stress fibers. However, if you for instance look at Fig.S9A, it is quite clear that cells expressing Rac1/F28L lack stress fibers.

2)        Considering the SD in Cdc42 12V expressing cells, it is unjustified to state that there is a clear shift towards filopodia relative to Cdc42 61L.

I have removed the statement.

3)        It may be due to the poor image quality in the pdf but no clear filopodia protrusions are apparent in the representative images for Cdc42Q61LT35A overexpression. The same applies for Cdc42Q61L delta ins.

I agree with the reviewer that the quality of the images were low. I have replaced these images with new images to support this statement.

4)      Immunoprecipitation:

a.       expression levels of the different constructs vary considerably and the amount of precipitated GTPases was not assessed in the IP WB.A reprobing of the membranes with an anti-myc Ab from a different species would provide this information allowing comparison of co-IPed GDI/mutant GTPase amounts

See below under 4c

b.      Comparing the IPs with the expression levels in the TCL WB, it appears that Q61L already has a much decreased affinity to GDI possibly, similar to that of Q61LR66A. Interestingly, with the possible exception of the insert region deletion, all other co-mutations appear to increase Q61L affinity for RhoGDI. This experiment needs to be properly clarified.

See below under 4c

c.       Overall, the experiments of interfering with RhoGDI binding were inconclusive and their pertinence is unclear, since the binding of RhoGDI is impaired in GTP-bound GTPases.

I have decided to remove the WB data. It does not add to the present study. It is already known that the R66A and deltainsert mutants do not bind RhoGDI. Measuring the RhoGDI binding constants for the different Cdc42 mutants have to be done in a focused study using different methods, not only immunoprecipitation. However, this is clearly outside the scope of the present study.

4)        Overall, the experiments with prenylation inhibitors and CAXX mutants do not add to previous knowledge.

The experiments were performed as controls in this side-by-side analysis. Most of the data is included in the supplementary section. However, Reviewer #4 requested a cells shape analysis of the CAAX box mutants so this has been performed and presented in the revised Fig.3D-F.

5)        Experiments with Rac1B strongly suggest that at least some of the structures induced by overexpressing fast-cycling mutants and identified as filopodia may actually be overexpression artifacts, since Rac1b has been shown to fail to signal to the cytoskeleton rearranging machinery – also seen here through the lack of response upon overexpression of the "classical" constitutively active Q61L mutant.

I am not sure if this has been established beyond all doubts. It is true that Peter Jordan (Matos et al. JBC 278, 50442) found that Rac1B and Rac1B/Q61L does not induce lamellipodia but have not found any information regarding filopodia. I clearly detect filopodia in Rac1B-expressing cells.

8)        The discussion is largely speculative and should be rewritten to focus on the relevant cytoskeletal changes observed after cellular integrity/viability and construct expression levels are accounted for.

I have tried to balance the rather diverse recommendations from reviewer #2 and #3 in order to make it less speculative but also more relevant.

Minor comments:

1)        The sentence “A large number of studies on Cdc42 and Rac1 have used the G12V and Q61L mutants of these GTPases. These mutations are known to block the hydrolysis activities of the small GTPases (14). It is therefore difficult to understand why they should be considered constitutively active.” is provocative but its reach has to be better explained, otherwise may seem paradoxical to the readers.

I have removed this statement.

2)        “Importantly, the Rac1B variant turned out to have much higher intrinsic GDP/GTP exchange activity, and might therefore qualify as a fast-cycling mutant protein (20,21).” ; “Rac1B is a cancer-associated fast-cycling splice variant of Rac1 (20,21).”; Work from Peter Jordan’s lab on Rac1b higher GDP/GTP exchange activity  in vivo proceeds that of the cited reports (J Biol Chem. 2003, 278(50): 50442-8) and should be mentioned.

I am sorry that I forgot to mention this. The reference is discussed and included in the revised manuscript.

3)        Fig S3- no A panel; unclear which mutant is represented in the top images; cells expressing low and High GTPase levels show completely different phenotypes.

This is now S4 and I have removed “A”. However, I do not agree that there are different phenotypes, both cells lack stress fibers but the right-hand cells appears to lack filopodia (which is true for 30% if the Cdc42/R66A-expressing cells)

4)        Line 323 – “… from 89& to…”.

5)         

This has been corrected to 89%

Reviewer 4 Report

The function of the Rho GTPases Rac1 and Cdc42 were in the past and are still analysed by overexpression of wild type Rac1 and Cdc42 and overexpression of mutant forms, which are locked in the GTP-bound form or have a high preference for the GTP-bound form, assuming that they all will show qualitatively similar responses. Yet, this might not be true and for example differential presence of fast-cycling and GTPase deficient Rac1 mutants in cancer indicates pathologically relevant differences between their functionality.

This manuscript by Pontus Aspenström analyses the effects of a number of Cdc42 and Rac1 mutants on the actin cytoskeleton and morphology of a fibroblastoid cell line. This is an important work as it challenges long established conceptions and highlights that we have to be more careful in the interpretation of the results obtained by overexpression of Cdc42 and Rac1 mutants. Despite shortcomings with respect to the biochemical analysis, this work gives a fresh look on how to analyse Rho GTPase function and will trigger important discussions.

Major comments:

1. The observed changes in actin cytoskeleton and cell morphology might be qualitatively dependent on the amount of the expressed Rac1 or Cdc42 mutant. Could it for example be that a low amount of GTPase deficient mutant will result in the same changes as a very high amount of wt or a medium amount of fast cycling mutant? Or are there qualitatively different changes between them independent of the expression level suggesting that the different forms have altered interactions with effectors? GEF dependent activation of wt and to a lesser extent of fast cycling mutants might facilitate interaction with certain effectors, while GTPase deficient mutants probably lack these preferential interactions.

To check the role of different amounts of Rho GTPase mutants, also cells with lower transfection levels should be analysed quantitatively. Maybe the morphological data of wt, fast cycling and GTPase deficient mutants (without additional point mutations or treatment with inhibitors) can be correlated with the quantified cellular fluorescence for the transfected Rho GTPases to assess whether differential expression levels lead to qualitatively different changes in actin cytoskeleton and cell morphology. 

2. The quantitative analysis of the fluorescent pictures is crucial for the understanding and should be part of the Methods section.

3. All findings should be put in a scheme indicating which mutant is involved in which alteration of the cytoskeleton or cell morphology and which effector pathways might be involved (inhibitor experiments).

Minor comments:

1. Line 62: “It is therefore difficult to understand why they should be considered constitutively active”. These mutants are called of course constitutively active because the GTP-bound form is considered to be the “active” form able to interact with effectors mediating the biological effects, as also mentioned by the author in line 54. Therefore, the sentence in line 62 should be deleted.

2. Filopodia are morphologically only by time lapse movies distinguishable from retraction fibers. This should be considered in the description of the results and the interpretation.

3. Figures should be provided at higher resolution. Especially the quantifications are hard to read.

4. “Significant” changes are mentioned several times but there is no information in the Methods about how was significance was calculated.

5. Arrowheads in figures should be described in the legends. 

6. Cdc42T17N expression results in more filopodia than mock transfected cell. Is this a real difference? It is also not mentioned in the results.

7. Clear information should be given in the Methods part about the tags used in each construct.

8. The elongated phenotype of the Cdc42 SAAX mutants is interesting and should be quantified.

9. “TCL” in Fig. 3D probably stands for total cell lysate, but the unexplained abbreviation should better be replaced as it is also used to the Rho GTPase TCL.

10. The prenylation inhibitor results are differentially described in the results and the discussion section (Discussion: “This treatment resulted in a relatively severe cellular phenotype, which included the loss of adhesion and stress fibers, and thus the effects on stress fiber dissolution could not be explored” vs. results “did not influence basic cellular responses” “did not visible affect cytoskeletal reorganization”). All results should be described completely in the results.

11. The discussion wrongly states that Cdc42 is upstream of Rac1 (Line 344). This should be corrected.

Author Response

Reviewer #4

Comments and Suggestions for Authors

The function of the Rho GTPases Rac1 and Cdc42 were in the past and are still analysed by overexpression of wild type Rac1 and Cdc42 and overexpression of mutant forms, which are locked in the GTP-bound form or have a high preference for the GTP-bound form, assuming that they all will show qualitatively similar responses. Yet, this might not be true and for example differential presence of fast-cycling and GTPase deficient Rac1 mutants in cancer indicates pathologically relevant differences between their functionality.

This manuscript by Pontus Aspenström analyses the effects of a number of Cdc42 and Rac1 mutants on the actin cytoskeleton and morphology of a fibroblastoid cell line. This is an important work as it challenges long established conceptions and highlights that we have to be more careful in the interpretation of the results obtained by overexpression of Cdc42 and Rac1 mutants. Despite shortcomings with respect to the biochemical analysis, this work gives a fresh look on how to analyse Rho GTPase function and will trigger important discussions.

Major comments:

1. The observed changes in actin cytoskeleton and cell morphology might be qualitatively dependent on the amount of the expressed Rac1 or Cdc42 mutant. Could it for example be that a low amount of GTPase deficient mutant will result in the same changes as a very high amount of wt or a medium amount of fast cycling mutant? Or are there qualitatively different changes between them independent of the expression level suggesting that the different forms have altered interactions with effectors? GEF dependent activation of wt and to a lesser extent of fast cycling mutants might facilitate interaction with certain effectors, while GTPase deficient mutants probably lack these preferential interactions.

There is presumably a difference between the GTPase deficient and fast-cycling mutants in their binding preferences for regulators and effectors. We know for instance that RhoGAP (Cdc42-GAP) cannot stimulate the hydrolysis activity of Cdc42Q61L mutant, but it can activate the GTPase activity of Cdc42/F28L. The differences in the sensitivity to the formin and Arp2/3 inhibitors between Q61L and F28L mutants also argue for differences in effector preferences between the various categories of mutant Cdc42 and Rac1.    

To check the role of different amounts of Rho GTPase mutants, also cells with lower transfection levels should be analysed quantitatively. Maybe the morphological data of wt, fast cycling and GTPase deficient mutants (without additional point mutations or treatment with inhibitors) can be correlated with the quantified cellular fluorescence for the transfected Rho GTPases to assess whether differential

I have aimed, throughout the study, to quantify cells that express the Cdc42 and Rac1 variants at the similar expression level. The criteria for quantifications are shown and discussed in the revised supplementary Fig2.S1 and S2. See comments to Reviewer #1-#3. 

2. The quantitative analysis of the fluorescent pictures is crucial for the understanding and should be part of the Methods section.

I have included this information in the revised Experimental section.

3. All findings should be put in a scheme indicating which mutant is involved in which alteration of the cytoskeleton or cell morphology and which effector pathways might be involved (inhibitor experiments).

I have included a summary of this information in a new table (table I) 

Minor comments:

Line 62: “It is therefore difficult to understand why they should be considered constitutively active”. These mutants are called of course constitutively active because the GTP-bound form is considered to be the “active” form able to interact with effectors mediating the biological effects, as also mentioned by the author in line 54. Therefore, the sentence in line 62 should be deleted.

I have deleted the sentence.

2. Filopodia are morphologically only by time lapse movies distinguishable from retraction fibers. This should be considered in the description of the results and the interpretation.

I have included comments about this fact in the revised manuscript.

3. Figures should be provided at higher resolution. Especially the quantifications are hard to read.

I agree totally with the reviewer. I have increased the size of the embedded images and text and submitted high-resolution revised figures.

4. “Significant” changes are mentioned several times but there is no information in the Methods about how was significance was calculated.

Significance determinations have been included in the revised text. Significance has only been quantified in Fig.3D-F.

5. Arrowheads in figures should be described in the legends. 

This has been explained in the revised figure legends.

6. Cdc42T17N expression results in more filopodia than mock transfected cell. Is this a real difference? It is also not mentioned in the results.

Cdc42T17N expression appear to cause increased filopodia formation compared to mock transfected cells, however the differences is not significant (p=0.0813). Therefore, this is discussed in the text. 

7. Clear information should be given in the Methods part about the tags used in each construct.

This information is provided in the revised Experimental section.

8. The elongated phenotype of the Cdc42 SAAX mutants is interesting and should be quantified.

This has been quantified and the data is included in the revised Fig.3D-F. The SAAX mutants are significantly more elongated.

9. “TCL” in Fig. 3D probably stands for total cell lysate, but the unexplained abbreviation should better be replaced as it is also used to the Rho GTPase TCL.

The Western blot data has been removed in the revised version of the article.

10. The prenylation inhibitor results are differentially described in the results and the discussion section (Discussion: “This treatment resulted in a relatively severe cellular phenotype, which included the loss of adhesion and stress fibers, and thus the effects on stress fiber dissolution could not be explored” vs. results “did not influence basic cellular responses” “did not visible affect cytoskeletal reorganization”). All results should be described completely in the results.

I understand that this appears confusing and I have included I more thorough description in the revised text.

11. The discussion wrongly states that Cdc42 is upstream of Rac1 (Line 344). This should be corrected.

This has been corrected

Round  2

Reviewer 1 Report

To this reviewer, new Figure-S2 clearly indicates that the author recognized a wide range of expression levels to be equal. When I commented “all microscopic examinations in this study needed extremely careful selection of cells …”, I expected an experiment such as the one suggested in Major concern 3 by Reviewer #3. Therefore, the author’s response in the revised manuscript is far from satisfactory. In addition, the unequivocal criteria to judge too high or too low expression levels were not described in the revised manuscript. Did the author quantify expression levels of Cdc42 and Rac1 in each cell throughout images and apply certain thresholds?

    If the current set of data cannot exclude the possibility that Cdc42/F28L but not Cdc42/Q61L is repressed by Cdc42-GAP, the author cannot tell whether the difference between F28L and Q61L is qualitative (different binding preferences for regulators and effectors) or quantitative (different levels of GTP-bound, active form of Cdc42). The author is clearly biased to the former, which is exemplified in the reply to Major comment 1 of Reviewer #4. But if the latter is the case, Cdc42/F28L simply behaved like wild type Cdc42 in the author’s experiments; the activity (the amount of GTP-bound active form) of Cdc42 rather than the intrinsic GDP/GTP exchange activity is a critical determinant on F-actin structure. Therefore, the presented data set does not appear adequately supporting the authors’ conclusion stated in the title.

Author Response

To this reviewer, new Figure-S2 clearly indicates that the author recognized a wide range of expression levels to be equal. When I commented “all microscopic examinations in this study needed extremely careful selection of cells …”, I expected an experiment such as the one suggested in Major concern 3 by Reviewer #3. Therefore, the author’s response in the revised manuscript is far from satisfactory. In addition, the unequivocal criteria to judge too high or too low expression levels were not described in the revised manuscript. Did the author quantify expression levels of Cdc42 and Rac1 in each cell throughout images and apply certain thresholds?

I can understand the concern but the important matter is that I have done the quantification by looking at the cells through the microscope. The reason for this experimental design is that it allows a thorough analysis of the reorganization of the actin filament system triggered by the different Cdc42 and Rac1 variants. It also allows for the analysis of a large number of cells within a reasonable time. Looking at acquired images is in this regard suboptimal since you often need to change focus and/or magnification in order to get a deeper understanding of what is going on with the actin filaments. Analyzing tens of thousands of cells would not have been possible without the analysis directly by the microscope. I cannot see how the analysis of F-actin organization (which is visible at the TRITC channel) could be a question of exposure time, since you analyze actin organization rather than intensity. The signal strength on the FITC channel (Myc-tagged Cdc42 and Rac1) is off course an important matter and you really need to have control over the situation. I can understand that my statements sound unscientific; what is too much and what is too little expression? However, when you look in the microscope, it immediately becomes quite clear which cells to avoid scoring. In my rebuttal letter, I have enclosed an example (Figure C) of what I think could serve as threshold values. As you can see, a “medium” intensity gives rise to the same phenotypic effect as cells with a “lower” or “higher” intensity (in this case, the cells have been exposed the same way). More signal on the FITC signal will not result in more TRITC signal, etc. Thus, the phenotypic response visible in the TRITC channel is not a matter of exposure time.  Moreover, I am not sure if a stronger intensity in the green channel always means higher expression of Myc-tagged Cdc42/Rac1. I think signal strength is also a matter of subcellular localization exogenous proteins and this is something that most probably will affect the accessibility of the Myc epitope. Thus, linking staining intensity is a complex matter and probably not as straight forward as you would like. I do not pretend that my experimental design is without flaws, but all variants of Cdc42 and Rac1 have been transfected in the same cell system for exactly the same time-period, and the slides were stained and handled exactly the same way.  I reasoned that this design should allow for a thorough side-by-side analysis in order to identify differences between the various mutants.

    If the current set of data cannot exclude the possibility that Cdc42/F28L but not Cdc42/Q61L is repressed by Cdc42-GAP, the author cannot tell whether the difference between F28L and Q61L is qualitative (different binding preferences for regulators and effectors) or quantitative (different levels of GTP-bound, active form of Cdc42). The author is clearly biased to the former, which is exemplified in the reply to Major comment 1 of Reviewer #4. But if the latter is the case, Cdc42/F28L simply behaved like wild type Cdc42 in the author’s experiments; the activity (the amount of GTP-bound active form) of Cdc42 rather than the intrinsic GDP/GTP exchange activity is a critical determinant on F-actin structure. Therefore, the presented data set does not appear adequately supporting the authors’ conclusion stated in the title.

I am not claiming that the increased GDP/GTP activity is the only decisive factor, it is most likely a mixture of regulatory proteins and effectors, and the intrinsic kinetic properties of the mutants, and the one does not exclude the other. I sure that there are critical differences in the effector binding abilities and I mention this as a possibility for the differences between the various mutants. However, in order to tell which regulators and which effectors would require a new and focused investigation. I can see this as a logical next step, but it is outside the scope of this article.

Reviewer 2 Report

Although extensively changed, the MS still needs to be improved mainly in the section of Discussion, in order to further strengthen and demonstrate the significance and scientific merit of the study.  

Author Response

I have sincerely tried to improve the Discussion and I have removed the parts that could be considered speculative or out of the scope (as suggested by Reviewer #3). However, I would like to mention that the “Discussion” section is a rather subjective part of an article and should reflect, at least to some extent, the view of the author(s). Therefore, different readers will react very differently to the Discussion. This is probably the reason for the rather opposing comments from Reviewers #2 and #3. I have aimed to put the findings described in this article in relation to the current view of the field. I am not sure that I will be able to improve this part much more, unless reviewer #2 could be much more specific in his/her criticism.

Reviewer 3 Report

In this revised version, the author has been able to address satisfactorily several of the concerns raised by the original manuscript.

However, a major issue was somewhat disregarded by the author – the expression level of the mutant GTPases and how this can compromise evaluation of the described cellular effects.

The author has provided some DAPI stained images for the wt constructs but not for the mutant constructs. Stating that including the DAPI staining would over complicate the figure is not a solid argument. 

Notwithstanding, despite referring that the nuclear integrity was taken into account when selecting cells to be analyzed,  several of the cells indicated in Fig.S2, particularly those overexpressing Cdc42, show clear nuclear abnormalities, some even with partial chromatin condensation.

Also, how can the author be sure that the chosen "apparent expression level" is the best to document the phenotypic effects? What are the objective criteria supporting this assumption?  When is a cell considered to express too little of the mutant GTPase? And why?

Again, a proper phenotypic evaluation of the effects of the different GTPase mutants requires one of the scoring criteria to be the GTPase expression level.

I previously suggested low, intermedium and high expression intervals, based on signal intensity (assuming that all images were acquired with equivalent settings). However, after reading the comments by Reviewer #4, I agree that a better way to do this would be to properly quantify  the fluorescent signal intensity and correlate it to the morphological data.

The publication of the reported observations is only pertinent if the described phenotypic changes can be unequivocally associated to the signaling properties of the mutant GTPase and not the result of cellular artifacts caused by abnormal overexpression.

Author Response

In this revised version, the author has been able to address satisfactorily several of the concerns raised by the original manuscript.

However, a major issue was somewhat disregarded by the author – the expression level of the mutant GTPases and how this can compromise evaluation of the described cellular effects.

See below

The author has provided some DAPI stained images for the wt constructs but not for the mutant constructs. Stating that including the DAPI staining would over complicate the figure is not a solid argument. 

The DAPI images give a somewhat dirty impression due to the massive occurrence of aggregates associated with the jetPEI transfection method. This phenomenon often obscures the images showing DAPI staining. For that reason, I have taken photos on the green and red channel only. I have enclosed a couple of figures, called A and B, to exemplify this matter. Figure A demonstrates that there is no condensed nuclei, but a lot of jetPEI-associated aggregates.

Notwithstanding, despite referring that the nuclear integrity was taken into account when selecting cells to be analyzed,  several of the cells indicated in Fig.S2, particularly those overexpressing Cdc42, show clear nuclear abnormalities, some even with partial chromatin condensation.

It is already published that exogenous expression of Cdc42 and Rac1 can have an impact on nuclear integrity (see Muris et al., 2002. Constitutive active GTPases Rac and Cdc42 are associated with endoreplication in PAE cells. Eur. J. Cancer 38, 1775-1782). It is not uncommon to find multinucleated Cdc42-, and to some extent Rac1-expressing cells.

Also, how can the author be sure that the chosen "apparent expression level" is the best to document the phenotypic effects? What are the objective criteria supporting this assumption?  When is a cell considered to express too little of the mutant GTPase? And why?

Again, a proper phenotypic evaluation of the effects of the different GTPase mutants requires one of the scoring criteria to be the GTPase expression level.

I previously suggested low, intermedium and high expression intervals, based on signal intensity (assuming that all images were acquired with equivalent settings). However, after reading the comments by Reviewer #4, I agree that a better way to do this would be to properly quantify the fluorescent signal intensity and correlate it to the morphological data.

The publication of the reported observations is only pertinent if the described phenotypic changes can be unequivocally associated to the signaling properties of the mutant GTPase and not the result of cellular artifacts caused by abnormal overexpression.

I can understand the concern but the important matter is that I have done the quantification by looking at the cells through the microscope. The reason for this experimental design is that it allows a thorough analysis of the reorganization of the actin filament system triggered by the different Cdc42 and Rac1 variants. It also allows for the analysis of a large number of cells within a reasonable time. Looking at acquired images is in this regard suboptimal since you often need to change focus and/or magnification in order to get a deeper understanding of what is going on with the actin filaments. Analyzing tens of thousands of cells would not have been possible without the analysis directly by the microscope. I cannot see how the analysis of F-actin organization (which is visible at the TRITC channel) could be a question of exposure time, since you analyze actin organization rather than intensity. The signal strength on the FITC channel (Myc-tagged Cdc42 and Rac1) is off course an important matter and you really need to have control over the situation. I can understand that my statements sound unscientific; what is too much and what is too little expression? However, when you look in the microscope, it immediately becomes quite clear which cells to avoid scoring. In my rebuttal letter, I have enclosed an example (Figure C) of what I think could serve as threshold values. As you can see, a “medium” intensity gives rise to the same phenotypic effect as cells with a “lower” or “higher” intensity (in this case, the cells have been exposed the same way). More signal on the FITC signal will not result in more TRITC signal, etc. Thus, the phenotypic response visible in the TRITC channel is not a matter of exposure time.  Moreover, I am not sure if a stronger intensity in the green channel always means higher expression of Myc-tagged Cdc42/Rac1. I think signal strength is also a matter of subcellular localization exogenous proteins and this is something that most probably will affect the accessibility of the Myc epitope. Thus, linking staining intensity is a complex matter and probably not as straight forward as you would like. I do not pretend that my experimental design is without flaws, but all variants of Cdc42 and Rac1 have been transfected in the same cell system for exactly the same time-period, and the slides were stained and handled exactly the same way.  I reasoned that this design should allow for a thorough side-by-side analysis in order to identify differences between the various mutants.